# Improved Carrier-Based Modulation for the Single-Phase T-Type qZ Source Inverter

Vitor Fernão Pires [1,2,*], Armando Cordeiro [2,3], Daniel Foito [1,4], Carlos Roncero-Clemente [5], Enrique Romero-Cadaval [5] and José Fernando Silva [2,6]

[1]   EST Setubal, Instituto Politécnico de Setúbal, 2910-761 Setúbal, Portugal; daniel.foito@estsetubal.ips.pt
[2]   INESC-ID, 1000-029 Lisboa, Portugal; armando.cordeiro@isel.pt (A.C.);
      fernando.alves@tecnico.ulisboa.pt (J.F.S.)
[3]   ISEL—Instituto Politécnico de Lisboa, 1000-029 Lisboa, Portugal
[4]   CTS-UNINOVA and LASI, 2829-516 Monte da Caparica, Portugal
[5]   Department of Electric, Electronic and Automation Engineering, University of Extremadura,
      06006 Badajoz, Spain; carlosrc@unex.es (C.R.-C.); eromero@unex.es (E.R.-C.)
[6]   IST—Instituto Superior Técnico, Universidade de Lisboa, 1049-001 Lisboa, Portugal
[*]   Correspondence: vitor.pires@estsetubal.ips.pt

**Abstract:** The Quasi-Impedance-Source Inverter (Quasi-Z inverter) is an interesting DC-AC converter topology that can be used in applications such as fuel cells and photovoltaic generators. This topology allows for both boost capability and DC-side continuous input current. Another very interesting feature is its reliability, as it limits the current when two switches on one leg are conducting simultaneously. This is due to an extra conduction state, specifically the shoot-through state. However, the shoot-through state also causes a loss of performance, increasing electromagnetic interference and harmonic distortion. To address these issues, this work proposes a modified carrier-based control method for the T-Type single-phase quasi-Z inverter. The modified carrier-based method introduces the use of two additional states to replace the standard shoot-through state. The additional states are called the upper shoot-through and the lower shoot-through. An approach to minimize the number of switches that change state during transitions will also be considered to reduce switching losses, improving the converter efficiency. The proposed modified carrier-based control strategy will be tested using computer simulations and laboratory experiments. From the obtained results, the theoretical considerations are confirmed. In fact, through the presented results, it is possible to understand important improvements that can be obtained in the THD of the output voltage and load current. In addition, it is also possible to verify that the modified carrier method also reduces the input current ripple.

**Keywords:** impedance-source converters; multilevel T-type; shoot through; quasi-impedance-source inverter (qZS inverter)

## 1. Introduction

Standard two-level voltage source inverters (VSIs) are among the most popular power converters. Two-level bridge VSIs are widely used in a variety of applications around the world due to their numerous advantages, including high efficiency, low cost, and ease of implementation and operation. However, two-level VSIs also have some drawbacks. Their buck voltage characteristic is a significant disadvantage that may limit their use when the output voltage needs to be higher than the input voltage and a single stage power conversion is mandatory. Another disadvantage of two-level VSIs is the need for a dead time or dead band on each VSI leg to prevent transistor shoot-through (DC bus short-circuit).

Due to the limitations of two-level VSIs, several boost-capable inverters have been proposed. Inverters based on impedance-source networks have been successfully proposed.

The impedance-source inverter topology, which was originally defined by a two-port network consisting of four energy storage components feeding a bridge inverter, overcomes some problems of the two-level VSIs by exhibiting buck–boost characteristics [1–4]. Energy storage components connected to suitably driven VSIs can provide buck–boost voltages while also addressing the shoot-through issue. Shoot through is a topological restriction in VSIs, but in impedance-source inverters, it is used to obtain the boost voltage characteristics. This capability of using the shoot-through state provides an additional advantage in terms of reliability. Indeed, unlike in two-level VSIs, a noise-induced short-circuit between the VSI DC terminals does not damage the power semiconductors. This allows for the elimination of dead times, thereby improving waveform quality. Another advantage of using impedance-source networks is their integration compatibility with standard multi-level inverters. Due to these important features, new configurations of impedance-source networks have been proposed and developed [5], including networks based on active structures and transformer-based networks [6–8]. Among the several impedance source networks that have been proposed, the quasi-impedance-source (quasi-Z) inverter is one of the most widely used [9,10]. Because of their interesting capabilities, these converters can be used in a variety of applications, including photovoltaic generation and energy storage systems [10–14]. Other applications require a fault-tolerance capability since power device short-circuits do not cause immediate damage to the converter. On the other hand, due to their buck–boost, wide-ranging voltage, they allow, up to a certain extent, the reconfiguration of the circuit to maintain their operation without affecting the voltage and current waveforms severely [15–17]. Furthermore, these converters enable the use of PWM modulators which have been modified to account for the generation of the shoot-through time interval. This is true, for example, for Simple Boost Control (SBC), Maximum Boost Control (MBC), and Maximum Constant Boost Control (MCBC) [18].

The implementation of impedance source networks in multilevel inverters has also been quite successful. This is the case for the multilevel quasi-Z source (qZS) inverter, which employs well known topologies such as the neutral point clamped (NPC) or the T-Type [19–24]. This can be verified by several proposals for its use in renewable generators and other fault-tolerant applications. Regarding the application to renewable generators, they are very well adapted for photovoltaic applications. In fact, to achieve the necessary voltage level and enable the system to operate over a larger range of PV output voltages, a qZS inverter was taken into consideration for PV voltage regulation [25,26]. Their application in storage systems is another aspect in which these converters are well adapted, as presented by [27]. Electric vehicles are another application [28]. Another significant feature of impedance source networks is their ability to adapt to the most well-known multilevel PWM modulators with just minor modifications to produce the shoot-through state [29–33]. However, one issue with these topologies and modulators is that, during the shoot-through state, the output voltage changes immediately to zero. This normally results in decreased performance, increased electromagnetic interference, and harmonic distortion. Nonetheless, with a three-level inverter, a half DC-link short-circuit is feasible. This is due to the possibility of a short-circuit between the upper and middle DC terminals, as well as the lower and middle DC terminals. Thus, in addition to the standard shoot-through state, it is also possible to have two extra shoot-through states, namely upper shoot-through (UST) and lower shoot-through (LST), as described in [33]. Because of the potential benefits of UST and LST, this concept was incorporated into the three-level NPC Z-source and quasi-Z source inverters [34]. This concept was also applied to the three-phase T-Type Z-source and quasi-Z source inverters [35,36]. Aside from their application to three-phase topologies, the UST and LST concepts were also included in the Space Vector Modulation (SVM) method. Actually, apart from reference [36], all of the methods were developed only for the SVM PWM technique and were not designed to reduce the number of switching transitions.

As mentioned previously, using the upper and lower shoot-through states reduces electromagnetic interference (EMI) while decreasing the harmonic content of the load

voltage and current. Thus, the ability to implement this concept in the widely used single-phase quasi-Z source inverters could be a valuable asset. As a result, this paper presents a novel modulation strategy incorporating the UST and LST concepts for the multilevel quasi-Z source single-phase inverter. The proposed strategy is intended for use in known modulation strategies such as the SBC and MBC. The benefits of the proposed strategy will be stated, particularly in terms of load voltage and current harmonic distortion. Furthermore, it is possible to verify that the converter input current ripple will be reduced, while the converter efficiency will be improved. These aspects will be validated not only by simulation tests but also through some laboratory experiments. The article makes the following specific contributions:

- A new modulation strategy applied to the single-phase quasi-Z source three-level T-Type VSI is presented, allowing for an important improvement in the output current and voltage THD when compared with the existent modulation strategies;
- The introduction of a strategy to minimize the number of switching states during transitions without affecting the output voltage distortion, while decreasing switching losses;
- The new modulation strategy also reduces the ripple within the converter input current when compared with the existent modulation strategies;

In addition to this introduction, the rest of the paper is organized as follows: Section 2 presents the system under study, namely, the T-Type Single-Phase quasi-Z source inverter. This includes the topology, an analysis of the circuit, and the design of its components. Next, Section 3 describes the newly proposed modulation strategy proposed for the T-Type Single-Phase quasi-Z source inverter. This strategy has the purpose of improving the output AC voltage and current THD, as well as the efficiency of the converter. Section 4 presents a discussion and a comparison of the proposed improvements considering the current state-of-the-art research. Comparative simulation results between the classical and the proposed approach are presented in Section 5 to highlight the improvements achieved. In Section 6, several experimental results are presented, showing the practical implementation of the proposed modulation strategy. Final conclusions are reported in Section 7.

## 2. T-Type Single-Phase Quasi-Z Inverter

PV and fuel-cell applications require a converter to regulate DC voltages. In addition, when connecting to an AC grid, it is often necessary to increase the DC voltage. Another constraint for PV and fuel-cell applications is the requirement of a converter to consume a continuous input current. The quasi-Z source inverter is one of the converters capable of meeting these specifications. Figure 1 shows a single-phase converter using a bridge inverter and two additional inductors and capacitors [9].

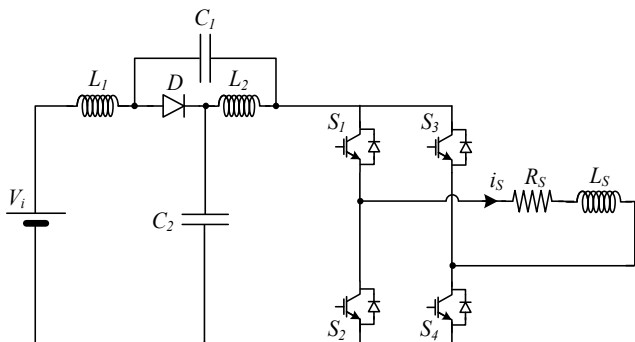

**Figure 1.** Single-phase two-level standard quasi-Z source inverter.

As previously stated, multilevel inverters can strongly improve the quality of AC voltages. Therefore, the quasi-Z impedance network was also included in multilevel inverters, such as the single-phase quasi-Z three-level T-Type VSI, which is one of the most

widely used and studied inverters (Figure 2). This converter requires the duplication of the impedance source network [22], and requires a total of four inductors and four capacitors.

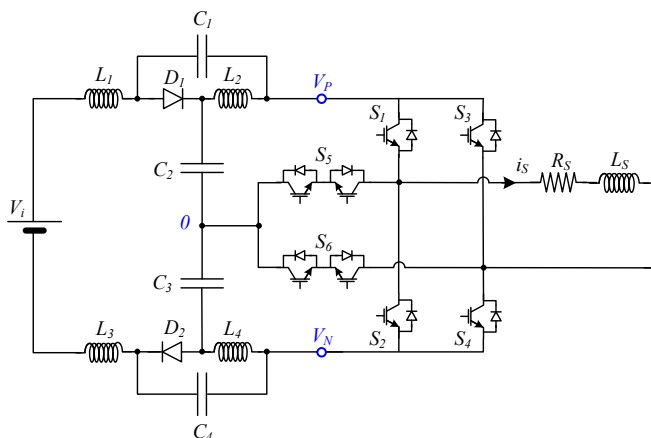

**Figure 2.** Single-phase three-level quasi-Z source inverter based on the T-Type configuration.

The analysis of the single-phase quasi-Z three-level T-Type VSI shows that three voltage levels ($+V_{PN}/2$, 0 and $-V_{PN}/2$) can be applied to the AC output. In addition to these three voltage levels determined by the combination of the switches in the standard VSI, the impedance network introduces an additional zero state caused by the short-circuit introduced by turning on simultaneously the upper and lower switches. Thus, the converter operation can be viewed as the result of the two equivalent circuits as depicted in Figure 3. The first mode happens when the inverter is in standard operation (without the shoot-through state), while the second mode happens when a leg is driven into the short-circuit state (shoot-through). From these two equivalent circuits, the converter analysis can be accomplished. considering that all components are ideal. In this way, applying Kirchhoff's laws to the two circuits, the AC output voltage can be calculated using (1) and (2) [37].

$$
\begin{cases}
v_{L1} = \dfrac{V_i}{2} - V_{C2}, \; v_{L2} = -V_{C1} \\
v_{L3} = \dfrac{V_i}{2} - V_{C3}, \; v_{L4} = -V_{C4} \\
v_{PN} = V_{C1} + V_{C2} + V_{C3} + V_{C4}
\end{cases}
\tag{1}
$$

$$
\begin{cases}
v_{L1} = \dfrac{V_i}{2} + V_{C1}, \; v_{L2} = V_{C2} \\
v_{L3} = \dfrac{V_i}{2} + V_{C4}, \; v_{L4} = V_{C3} \\
v_{PN} = 0
\end{cases}
\tag{2}
$$

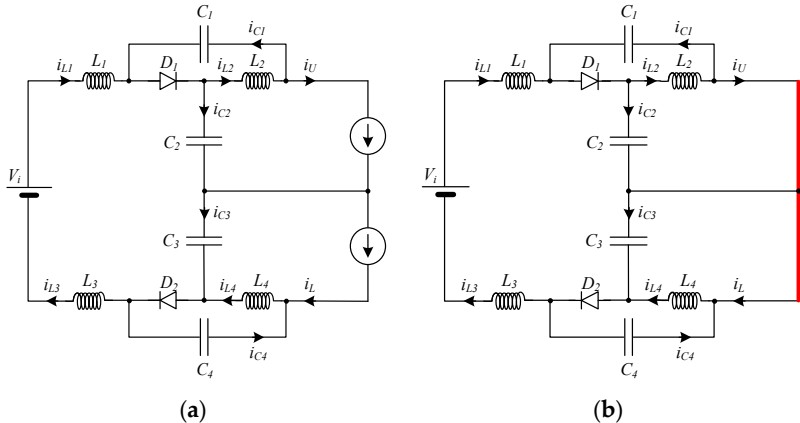

**Figure 3.** Equivalent circuits associated with the two circuit states (**a**) non shoot-through state (**b**) shoot-through state represented by a short-circuit in red color.

Equations (1) and (2) can be used to calculate the static voltage gain achieved by using the impedance network. The average voltage of the capacitors can be determined using the above equations and can be written as:

$$\begin{cases} V_{C1} &= V_{C4} &= \frac{D_S}{2-4\,D_S}V_i \\ V_{C2} &= V_{C3} &= \frac{1-D_S}{2-4\,D_S}V_i \end{cases} \tag{3}$$

where $D_S$ is the duty cycle's shoot-through state.

Using the capacitors' average voltages expressed in (3), the static voltage gain given by the impedance network associated with the T-Type inverter can be calculated as:

$$V_{PN} = V_{C1} + V_{C2} + V_{C3} + V_{C4} = \frac{1}{1-2\,D_S}\,V_i \tag{4}$$

As expected, the quasi-Z three-level T-Type VSI provides a voltage gain that is crucial for the aforementioned applications.

The inductance and capacitance values of a quasi-Z three-level T-Type VSI are calculated considering a specific maximum current and voltage ripple under nominal operation. The current and the voltage ripples $\Delta i_L$ and $\Delta V_C$, respectively, can be obtained from the approximated dynamic behavior of the inductor currents and capacitor voltages. The next equations are adopted for the upper quasi-Z components. Similar equations can be obtained for the lower quasi-Z components.

$$\frac{di_{L1}(t)}{dt} = \frac{v_{L1}(t)}{L_1} \tag{5}$$

$$i_{L1}(t) = \frac{v_{L1}}{L_1}t_1 + i_{L1}(t_o) \tag{6}$$

$$\frac{\Delta i_{L1}}{t_1} = \frac{v_{L1}}{L_1} \tag{7}$$

$$\frac{dv_{C1}(t)}{dt} = \frac{i_{C1}(t)}{C_1} \tag{8}$$

$$v_{C1}(t) = \frac{v_{C1}}{C_1}t_1 + v_{C1}(t_o) \tag{9}$$

$$\frac{\Delta V_{C1}}{t_1} = \frac{i_{C1}}{C_1} \tag{10}$$

From the previous relationships and the application of the Kirchhoff Laws, the passive parameters can also be obtained by:

$$L_1 = L_3 = \frac{t_1\left(\frac{V_i}{2}+V_{C_{1,4}}\right)}{\Delta i_{L_{1,3}}} = \frac{\delta(1-\delta)\,V_i}{2(1-2\delta)f_s\Delta i_{L_{1,3}}} \tag{11}$$

$$L_2 = L_4 = \frac{t_1 V_{C_{2,3}}}{\Delta i_{L_{2,3}}} = \frac{\delta(1-\delta)\,V_i}{2(1-2\delta)f_s\Delta i_{L_{2,3}}} \tag{12}$$

$$C_1 = C_4 = \frac{t_1 i_{C_{1,4}}}{\Delta V_{C_{1,4}}} = \frac{2\,\delta\,I_o}{(1-2\delta)f_s\Delta V_{C_1}} \tag{13}$$

$$C_2 = C_3 = \frac{t_1 i_{C_{2,3}}}{\Delta V_{C_{2,3}}} = \frac{(1-\delta)\,I_o}{f_s\Delta V_{C_{2,3}}} \tag{14}$$

where $I_o$ is the current flowing from the P terminal (see Figure 2) to the load and $t_1$ is the time associated with the shoot-through-state duty cycle.

The minimum current and voltage standing for the power semiconductors are determined from the assumption that the average values of the currents in the inductors

and voltages in the capacitors are zero at the end of the switching cycle ($T$). Therefore, the semiconductor's minimum standing current and voltage are:

$$I_{S_{1,\cdots,6}} = \left( \frac{4\delta}{1-2\delta} + \frac{1}{\delta} \right) I_o \tag{15}$$

$$I_{D_{1,2}} = \left[ \frac{4(1-\delta)}{1-2\delta} + \frac{1}{1-\delta} \right] I_o \tag{16}$$

$$V_{S_{1,\cdots,4}} = V_{PN} \tag{17}$$

$$V_{S_{5,6}} = \frac{V_{PN}}{2} \tag{18}$$

$$V_{D_1} = V_{C_1} + V_{C_2} = V_{D_2} = V_{C_3} + V_{C_4} = \frac{V_{PN}}{2} \tag{19}$$

## 3. Proposed Improved Sinusoidal Pulse-Width Modulation Strategy

One of the disadvantages of the qZS inverter and its existing modulation techniques is that the conventional shoot-through state degrades performance and increases electromagnetic interference and harmonic distortion. In fact, during the shoot-through phase, the AC voltage immediately drops down to zero. To mitigate these issues, a new modulation strategy is proposed here.

The proposed SPWM strategy for the qZS inverter starts with the existing Simple Boost SPWM concepts for impedance source inverters [38], including the modulator references for the generation of the shoot-through intervals, namely, the positive and negative shoot-through references ST_pos and ST_neg. Figure 4 illustrates the Simple Boost PWM strategy and its existing switching scheme. The shoot-through references are added to the SPWM strategy with a value of $\pm(1-D_S)$ to provide the desired static voltage gain as expressed in (4).

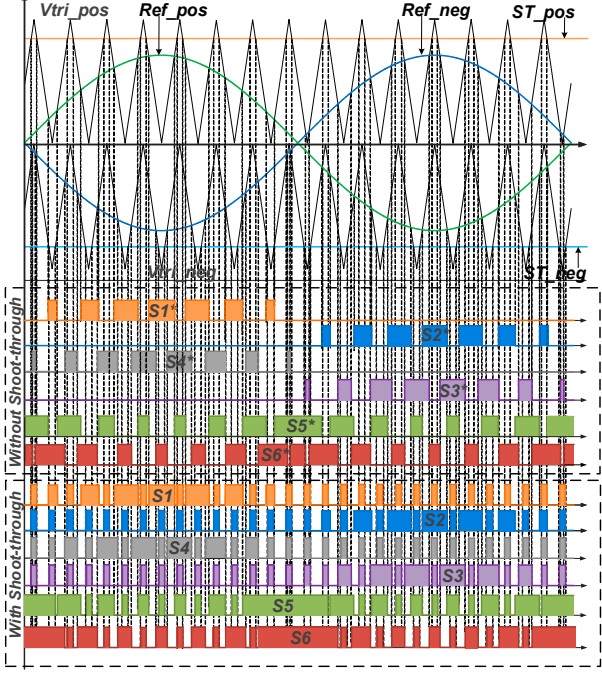

**Figure 4.** Modulators, carrier signals, and transistor gate signals for the existing Boost SPWM modulation strategy. The S1*–S6* are the transistor gate signals without shoot-through and the S1–S6 are the transistor gate signals considering the shoot-through.

In the existing Simple Boost SPWM modulator, the shoot-through state is associated with a short-circuit in the full DC link (Figure 3b) due to connecting the three terminals, P,

O, and N, together directly. However, in the proposed Improved Boost SPWM modulation there are two additional switching states associated with these new shoot-through states. One of the shoot-through states is available through connecting the P and O terminals together (Figure 5a), which is called the upper shoot-through (UST). The second shoot-through state is available by short-circuiting the O and N terminals (Figure 5b), and is called the lower-shoot-through (LST). Using the additional UST and LST states has the benefit of allowing the inverter output to be powered by the intermediate DC voltage even during the shoot-through state.

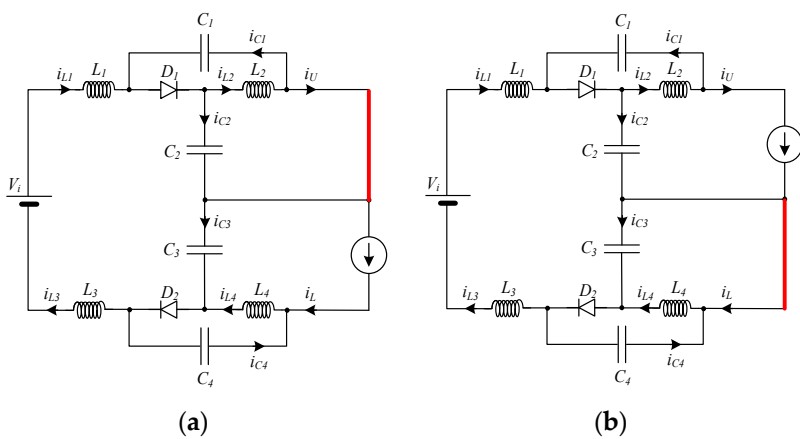

(**a**)       (**b**)

**Figure 5.** Equivalent circuits associated with the two extra shoot-through states: (**a**) upper shoot-through state, (**b**) lower shoot-through state. Shoot-through states (or short-circuit states) are represented in red color.

To use the two extra shoot-through states in the new SPWM modulation strategy, Table 1 shows an analysis of the possible voltage level functions of the switches' states (voltage level is $V_{PN}/2$). In contrast to the existing SPWM, this table takes into account both the UTS and LST states. It also considers the voltage level that changes the least during the shoot-through state. This table also indicates that there are several redundant states.

**Table 1.** Possible voltage levels function of the switches' states, showing the switches states, corresponding voltage level available in the upper shoot-through (UTS) and lower shoot-through (LTS). The necessary changes in the power switches are also described for each voltage level.

| Voltage Level | State of the Switches | | | | | | Voltage Level with UTS or LTS ON | Switches Changing State with UTS or LTS ON |
|---|---|---|---|---|---|---|---|---|
| | S1 | S2 | S3 | S4 | S5 | S6 | | |
| +2 V | 1 | 0 | 0 | 1 | 0 | 0 | $+V_{PN}/2$ | S5 |
| | 1 | 0 | 0 | 1 | 0 | 0 | $+V_{PN}/2$ | S6 |
| +V | 1 | 0 | 0 | 0 | 0 | 1 | $+V_{PN}/2$ | S4, S5, S6 |
| | 0 | 0 | 0 | 1 | 1 | 0 | $+V_{PN}/2$ | S1 |
| 0 | 1 | 0 | 1 | 0 | 0 | 0 | 0 | S2, S4, S5, S6 |
| | 0 | 1 | 0 | 1 | 0 | 0 | 0 | S1, S3, S5, S6 |
| | 0 | 0 | 0 | 0 | 1 | 1 | 0 | S1, S2, S3, S4 |
| −V | 0 | 0 | 1 | 0 | 1 | 0 | $-V_{PN}/2$ | S2 |
| | 0 | 1 | 0 | 0 | 0 | 1 | $-V_{PN}/2$ | S3, S5, S6 |
| −2 V | 0 | 1 | 1 | 0 | 0 | 0 | $-V_{PN}/2$ | S6 |
| | 0 | 1 | 1 | 0 | 0 | 0 | $-V_{PN}/2$ | S5 |

The states analysis presented in Table 1 allows for the development of the new PWM modulation strategy that takes into account the upper and lower shoot-through states in order to avoid decreasing more than one voltage level when the shoot-through is activated. Figure 6 shows the developed PWM modulation strategy along with the corresponding switching scheme. Figure 7 depicts the implementation of this strategy. This demonstrates that the implementation of this strategy is simple, as illustrated using digital circuitry. Another aspect to note in the comparison of the two strategies (Figures 4 and 6) is that the number of switches that change state during transitions is lower. This is a positive characteristic as it improves the converter's efficiency.

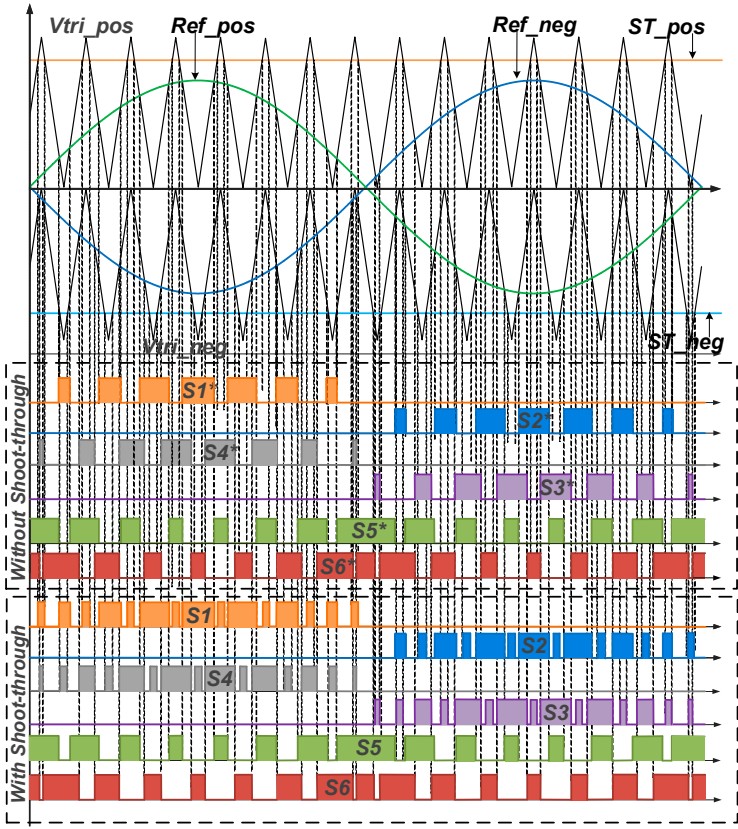

**Figure 6.** Modulating, carrier signals, and transistor gate signals for the proposed PWM modulation strategy. The S1*–S6* are the transistor gate signals without shoot-through and the S1–S6 are the transistor gate signals considering the shoot-through.

Additionally, the redundant states are used to reduce the number of switches that change state during transitions without affecting the output voltage distortion, while decreasing the switching losses. The idea is not to change the desired voltage level, but only to minimize the number of devices switching, maintaining in this way the THD of the output voltage and current. Based on the combinatorial analysis presented in Table 1, the machine state sequence obtained to reduce the number of switching devices is shown in Figure 8. In this diagram, the ellipses represent the state of the switches at that voltage level, while the arrows represent the switches that change state to reach a specific voltage level.

The developed transition scheme allows the closing approach for the newly proposed modulation strategy to be devised. Figure 9 depicts the new PWM modulation strategy and its corresponding switching scheme. This new scheme avoids the shoot-through states associated with the short-circuit between the upper and lower DC poles when a voltage level of zero is not desired. In fact, this strategy avoids lowering more than one voltage level when the shoot-through is activated. As will be shown, this results in a reduction

in the inverter output voltage THD. Figure 10 depicts the simple implementation of this strategy, illustrated using digital circuitry.

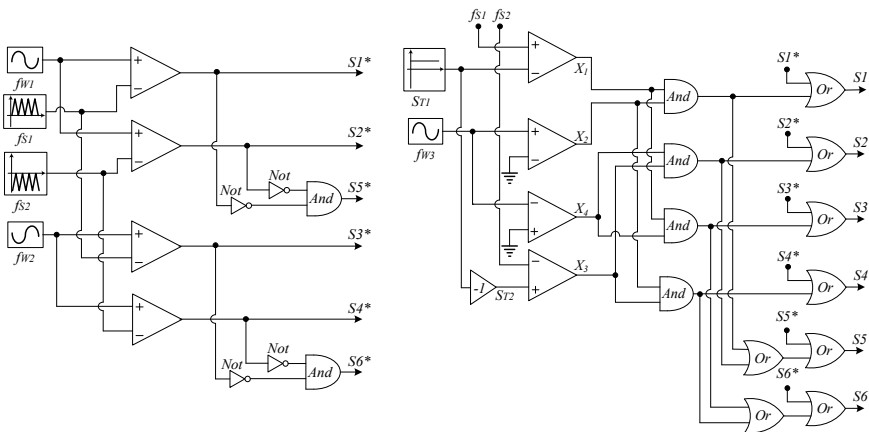

**Figure 7.** Implementation of the proposed PWM modulation strategy. The S1*–S6* are the transistor gate signals without shoot-through and the S1–S6 are the transistor gate signals considering the shoot-through.

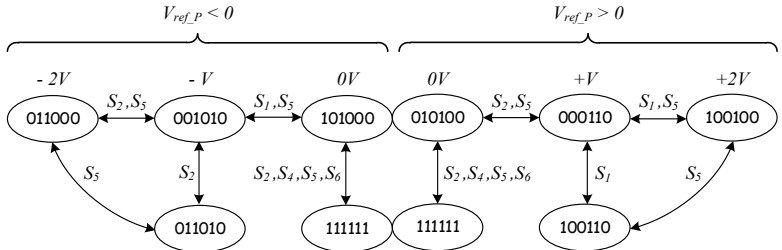

**Figure 8.** Switching combination leading to the minimization of the number of switches that change during the transitions.

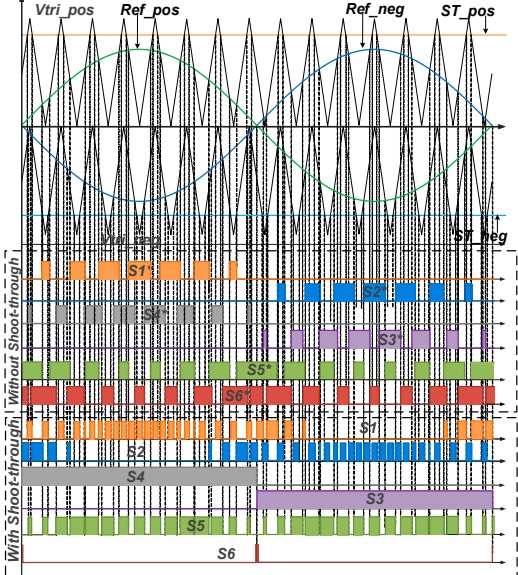

**Figure 9.** Modulators, carrier signals, and switch gate signals for the proposed SPWM modulation strategy with minimization of the number of switches that change during the transitions. The S1*–S6* are the transistor gate signals without shoot-through and the S1–S6 are the transistor gate signals considering the shoot-through.

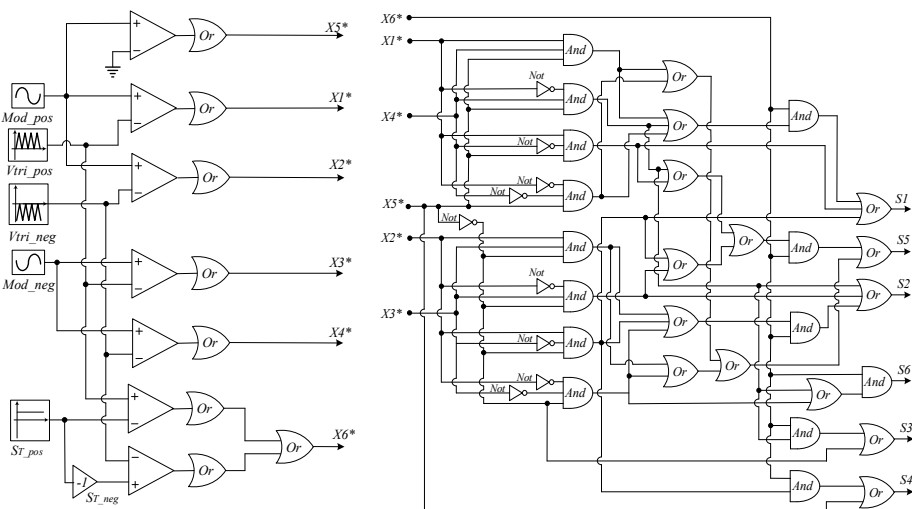

**Figure 10.** Implementation of the proposed SPWM modulation strategy with minimization of the number of switches that change during the transitions.

The new strategy here presented can be adapted to other modulation strategies, such as the MBC and the MCBC [38].

## 4. Comparative Study

This section presents, in Table 2, a comparison between the proposed modulation strategy for the T-Type single-phase quasi-Z inverter and some recent modulation strategies that, in addition to the standard shoot-through state, also use the two extra shoot-through states, namely upper shoot-through (UST) and lower shoot-through (LST).

**Table 2.** Comparison of the proposed modulation strategy with existing ones that also use the two extra shoot-through states.

| Topologies / Items | [31] | [32] | [33] | [34] | [Proposed] |
|---|---|---|---|---|---|
| Application to the single-phase inverter | No | No | No | No | Yes |
| Application to the three-phase inverter | Yes | Yes | Yes | Yes | Yes |
| Improvement of switches transitions | No | No | No | No | Yes |
| Base modulation strategy | SVPWM | SVPWM | SVPWM | SBPWM | SBPWM |
| Reducing the ripple of the converter input current | NA | NA | NA | NA | Yes |

NA—Not addressed.

Some recent modulation strategies are based on the traditional approach used in quasi-Z inverters, also using the two extra shoot-through states to improve the output voltage and current THD. However, as shown in Table 2, only the SPWM proposed in this paper can be applied to single-phase inverters. As a disadvantage, it cannot be applied to the three-phase inverter. This work is the first to propose a reduction in the number of switching states during transitions without affecting the output voltage distortion. Reducing the number of switching states reduces the switching losses and increases the converter efficiency. Furthermore, the proposed new SPWM modulation together with the strategy presented in [34] can be applied to the sinusoidal PWM modulation approach, whereas the remaining modulation strategies included in the comparison were developed based on the Space

Vector-based PWM technique. Further, the proposed new SPWM modulation also reduces the converter input current ripple, while the remaining modulation strategies included in the comparison do not seem to present this advantage.

## 5. Simulation Results

Computer simulation tests were performed to verify the described properties, behavior, and operation of the T-Type single-phase qZS inverter against the existing and the newly proposed modulation strategies. The presented computer simulation tests were obtained from the well-known and established program Matlab/Simulink R2018a using the Power Systems Toolbox. The circuit parameters used for these tests are shown in Table 3. These parameters were not chosen for a specific application, since the purpose was only to test the proposed modulation approach. However, those parameters that can be used in photovoltaic systems in which serial connections between PV panels are adopted. These values are also compatible with 120 V RMS low-voltage grids.

**Table 3.** Circuit parameters used for the simulation tests.

| Parameter | Value |
| --- | --- |
| Input *DC* voltage | 200 V |
| Capacitors $C_1$ and $C_4$ | 400 μF |
| Capacitors $C_2$ and $C_3$ | 1000 μF |
| Inductors $L_1$, $L_2$, $L_3$ and $L_4$ | 200 μH |
| Inductor Load $L_o$ | 10 mH |
| Resistor Load $R_o$ | 50 Ω |
| Switching frequency | 5 kHz |

The first simulation test was conducted using a modulation index and a shoot-through value of 0.8 and 0.05, respectively. This test was performed for both the existing and the newly proposed strategies. Figures 11–13 show the results of the AC output voltage, AC output current, and converter input current, respectively, comparing the two modulation strategies. Figure 11a shows the AC output load voltage of the existing modulation, defined as standard modulation, and Figure 11b shows the same voltage for the proposed new modulation. These figures show that the AC output voltage obtained using the newly proposed modulation strategy does not decrease by more than one voltage level when the shoot-through is activated. This reduces the output voltage THD, as this voltage tracks the desired reference more closely. This also leads to a reduction in the output current THD. Figure 12a shows the AC output current of the standard modulation and Figure 12b presents the same current for the proposed new modulation, considering the same modulation index and shoot-through value. The waveform in Figure 12b clearly suggests the current contains less distortion, as the current waveform (especially near zero crossing) is approximated to a sine reference. Another advantage of the newly proposed modulation strategy is the reduction in the converter input current ripple, as shown by Figure 13b, when compared to the standard modulation strategy, presented in Figure 13a. Indeed, the current ripple seen in the classical approach is about 53% against 41% for the newly proposed modulation strategy. One disadvantage of the newly proposed modulation strategy is that its converter boost capability is slightly reduced. In this case, the peak voltage reduction is about 17%. However, this reduction depends on the shoot-through duty cycle. Thus, for lower shoot-through duty cycle values, the reduction is even lower.

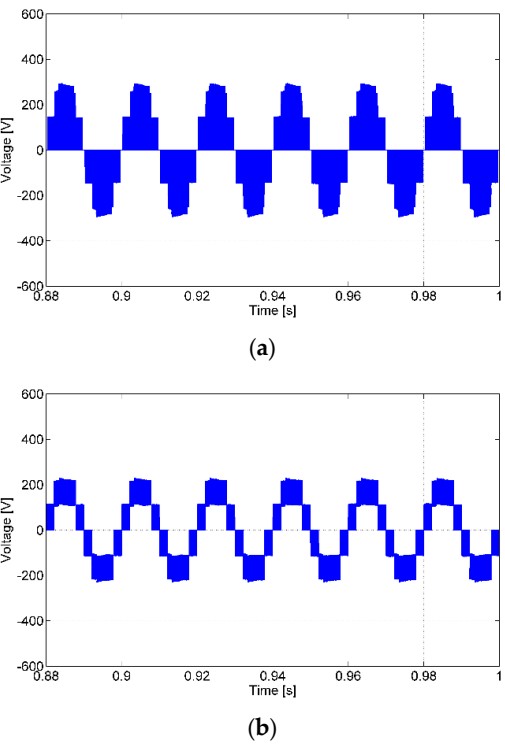

(**a**)

(**b**)

**Figure 11.** Simulation results of the load voltage for shoot-through of 0.05 and (**a**) standard modulation strategy or (**b**) newly proposed modulation strategy.

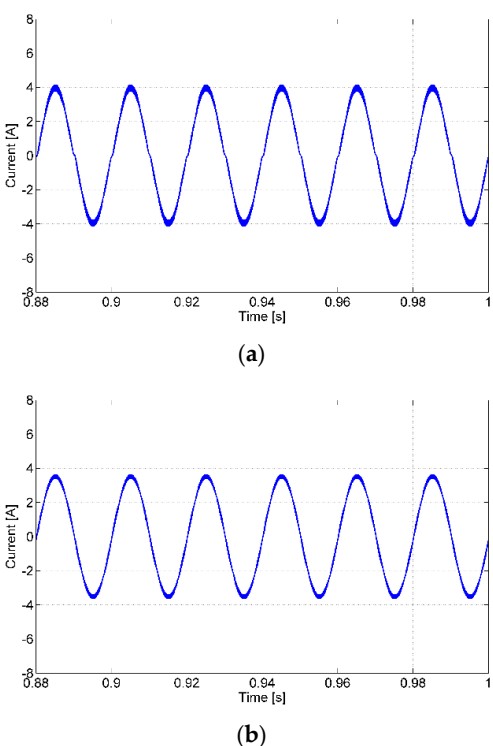

(**a**)

(**b**)

**Figure 12.** Simulation results of the load current for shoot-through of 0.05 and (**a**) standard modulation strategy or (**b**) newly proposed modulation strategy.

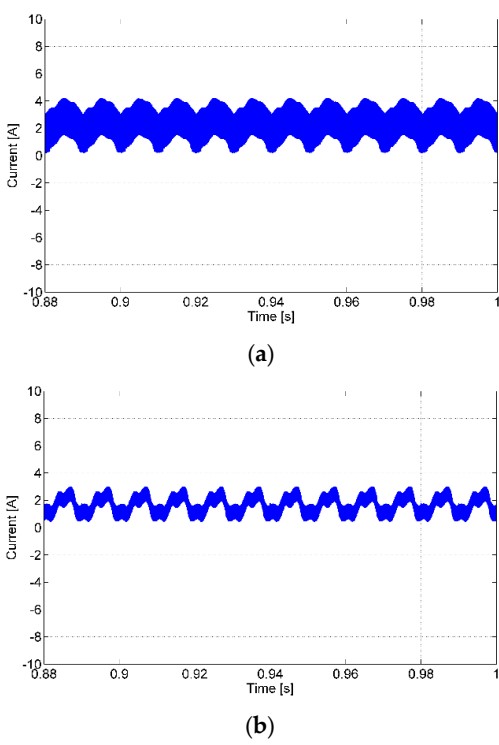

**Figure 13.** Simulation results of the converter input current for shoot-through of 0.05 and (**a**) standard modulation strategy or (**b**) newly proposed modulation strategy.

Simulation tests with an increased shoot-through value (but the same modulation index value of 0.8) for the standard and the newly proposed strategies were also performed. Figures 14–16 show the results of the AC output voltage, AC output current, and converter input current, respectively, comparing the two modulation strategies when the shoot-through increases to 0.08. As expected, the load voltage and current amplitudes increase, but the overall behavior is quite similar to the previous simulation tests. Figure 14a shows the AC output load voltage of the existing modulation, defined as standard modulation, and Figure 14b presents the same voltage for the newly proposed modulation, considering the same modulation index value of 0.8 and the new shoot-through value of 0.08. Again, this result shows that the new modulation generates a lower THD voltage. Figure 15a shows the AC output current waveform of the standard modulation and Figure 15b shows the same current for the proposed new modulation, considering the same modulation index and the new shoot-through value of 0.08. One noticeable difference is that the amplitude of the load current obtained with the newly proposed modulation strategy is now roughly equal to that obtained with the existing modulation strategy and shoot-through 0.05. One encouraging finding, though, is that the load voltage peak value obtained using the newly proposed modulation strategy and the shoot-through of 0.08 (Figure 14b) is smaller than the load voltage obtained using the standard modulation strategy (Figure 11a) with a shoot-through of 0.05. Therefore, the proposed strategy requires a lower load peak value voltage to achieve the same output load current. Indeed, the voltage peak in Figure 14b is 265 V (or 530 V peak-to-peak) and the peak voltage in Figure 11a is 300 V (or 600 V peak-to-peak). Several investigation studies show that the reduced peak voltage (direct and reverse voltage) decreases the probability of power devices' breakdown, decreases the losses, decreases the operation temperature, and increases the reliability of the power semiconductors [39]. The converter input current exhibits a similar behavior because the ripple is reduced with the newly proposed modulation strategy. In fact, Figure 16 shows that, with the newly proposed modulation, the input current ripple is smaller when compared with the classical modulation strategy. The current ripple value obtained with the classical approach is about 64% compared to 38% for the newly proposed modulation

strategy. So, considering the increased shoot-through value, the difference in current ripples obtained between the classical and the proposed strategies also increases. So, it is possible to conclude that this difference is not constant, being lower for lower shoot-through duty cycles.

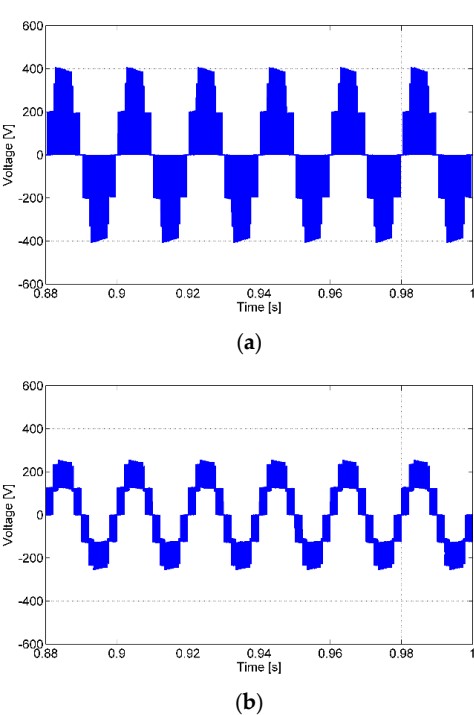

**Figure 14.** Simulation results of the load voltage for shoot-through of 0.08 and (**a**) standard modulation strategy or (**b**) newly proposed modulation strategy.

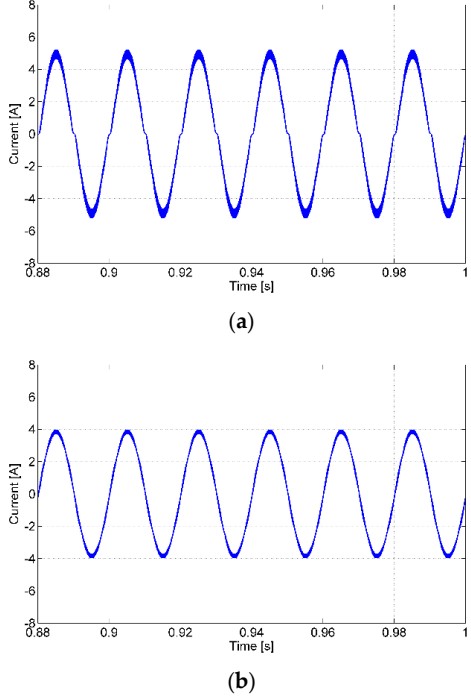

**Figure 15.** Simulation results of the load current for shoot-through of 0.08 and (**a**) standard modulation strategy or (**b**) newly proposed modulation strategy.

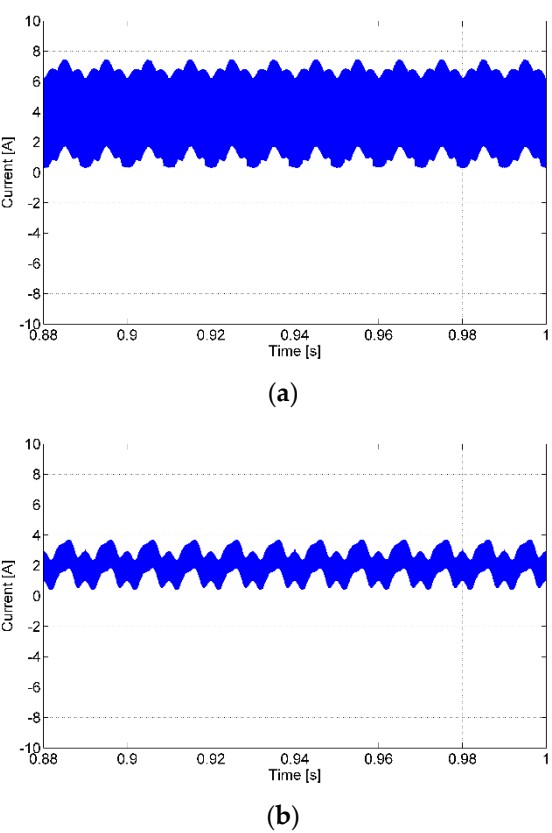

**Figure 16.** Simulation results of the converter input current for shoot-through of 0.08 and (**a**) standard modulation strategy or (**b**) newly proposed modulation strategy.

As shown by the previous results, the waveforms of the load voltage for the newly proposed modulation strategy are closer to the reference voltage. This can be confirmed by the analysis of the load voltage and current THD as functions of the shoot-through duty cycle. To confirm this behavior, several simulation tests were conducted for different shoot-through values for both modulation strategies. Figure 17 presents evolution graphs for the voltage and current THD as functions of the shoot-through duty cycle for the existing (standard) and newly proposed modulation strategies. Figure 17a shows a fast increase in the THDs of the load voltage (THD-V) and current (THD-I) waveforms as the shoot-through increases, when using the existing (standard) modulation strategy. This does not occur, however, with the newly proposed modulation strategy, which presents THD values that are nearly constant or slightly decreasing despite the increase in the shoot-through duty cycle (see Figure 17b). In this way, it is possible to conclude that the newly proposed modulation introduces a reduction of around 50% in the output voltage distortion and of around 13% in the output current.

As pointed out, the effect of the suggested modulation strategy on the converter efficiency is another crucial factor. The converter efficiency was calculated through additional simulation tests considering the existing and newly proposed modulation strategies. Figure 18 shows the efficiency of the converter as a function of the shoot-through duty cycle. This figure suggests that the converter efficiency increases by 1%, showing the real value and usefulness of the newly proposed modulation strategy.

Figure 19 presents the power-loss breakdown at the nominal power output of 950 W, considering the power semiconductors conduction losses, switching losses and passive components losses. From these results, one important aspect is that the switching losses associated with the newly proposed modulation strategy become smaller, as well as the conduction and passive components losses, when compared to those of the existing modulation, due to the lower harmonic content.

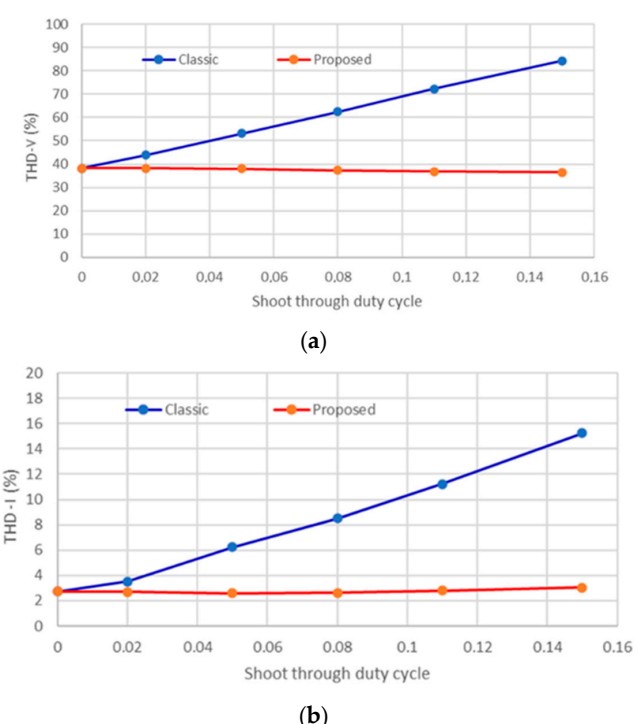

**Figure 17.** Simulation results of the THD as a function of the shoot-through duty cycle of the (**a**) load voltage (THD-V); (**b**) load current (THD-I).

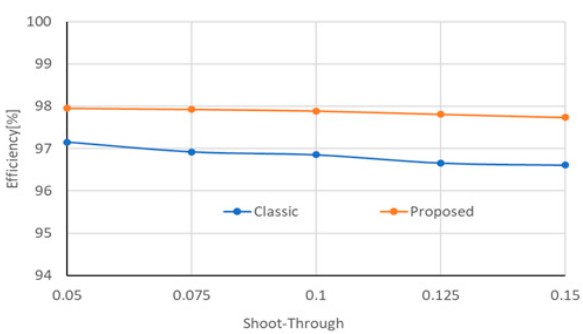

**Figure 18.** Simulation results of the efficiency as a function of the shoot-through duty cycle.

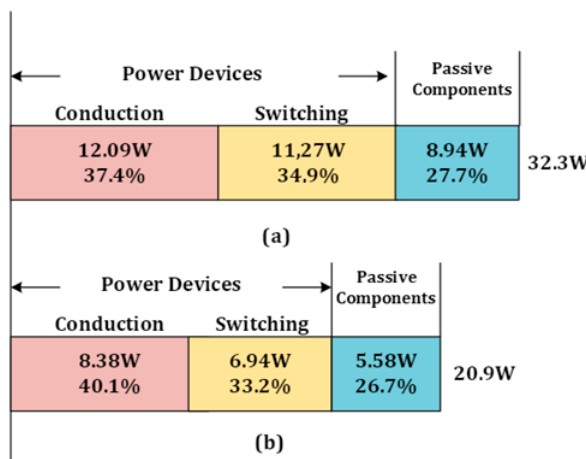

**Figure 19.** Power-loss breakdown at 200 V, 950 W and 5 kHz; (**a**)—classic modulation; (**b**) proposed modulation.

## 6. Experimental Results

A laboratory prototype was assembled in order to experimentally validate the obtained simulation results shown in the preceding section. The nominal values of the components of the converter laboratory prototype are matched to those used in the simulations (Table 3). Some details of this prototype developed to perform the experimental tests can be seen in Figure 20. This figure shows a general overview of the 950 W proposed prototype, where is possible to see (1) the auxiliary power source; (2) the single-phase full-bridge VSI (S1–S4) (IXA12IF1200HB Si IGBT); (3) the bidirectional power devices of the proposed topology (S5-S6) (IXA12IF1200HB Si IGBT); (4) the qZS circuit which includes two main electrolytic capacitors (C2 = C3 = 1000 µF), two auxiliary capacitor (C1 = C4 = 200 µF), and two diodes (DHG10I1200PM); (5) the four inductors used in the qZS circuit (L1 = L2 = L3 = L4 = 200 µH); (6) the gate drive circuit board for power devices; (7) the general purpose control board based on the Microchip DSP microcontroller 30F4012; (8) the main DC Power Source; (9) the load inductors; and (10) the load resistors.

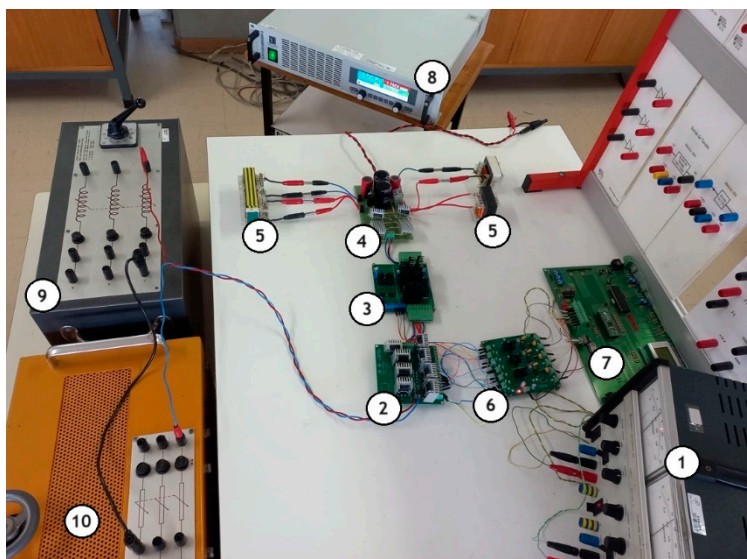

**Figure 20.** Experimental setup of the proposed system; (1)—the auxiliary power source; (2)—the single-phase full-bridge VSI; (3)—the bidirectional power devices of the proposed topology; (4)—the qZS circuit which includes two main electrolytic capacitors, two auxiliary capacitors, and two diodes; (5)—the four inductors used in the qZS circuit; (6)—the gate drive circuit board for power devices; (7)—the general purpose control board; (8)—main DC power source; (9)—load inductors; (10)—load resistors;.

The conducted experiments are also the same as in the simulation. Consequently, the initial experiment began with a shoot-through value of 0.05 and a modulation index of 0.8, and was performed using both the newly proposed and existing modulation strategies. The results of the AC output voltage, AC output current, and converter input current obtained as a result of these two modulation strategies are shown in Figures 21–23. The displayed results confirm how the proposed modulation strategy keeps the AC output voltage much closer to the reference by preventing the voltage levels from dropping more than one voltage level when the shoot-through is engaged (Figure 21). Figure 21 also shows that the newly proposed modulation generates a reduced voltage THD content which usually means less electromagnetic noise emission. The converter output current ripple is lower when using the newly proposed modulation technique when compared with the existing modulation technique (Figure 22). Figure 22 also shows that the newly proposed modulation generates a reduced current THD content which can be seen by the approximation to a perfect sine waveform. It is also possible to verify that, in contrast to the results obtained with the classical modulation strategy, the converter input current ripple obtained from the newly proposed modulation technique is smaller than that obtained

using the existing modulation strategy (Figure 23). Indeed, the current ripple value obtained with the classical approach is about 57%, while the current ripple value obtained with the newly proposed modulation strategy is about 46%. A reduction in the boost capability is also observed, which can be verified when using the newly proposed modulation strategy, which confirms the simulation tests.

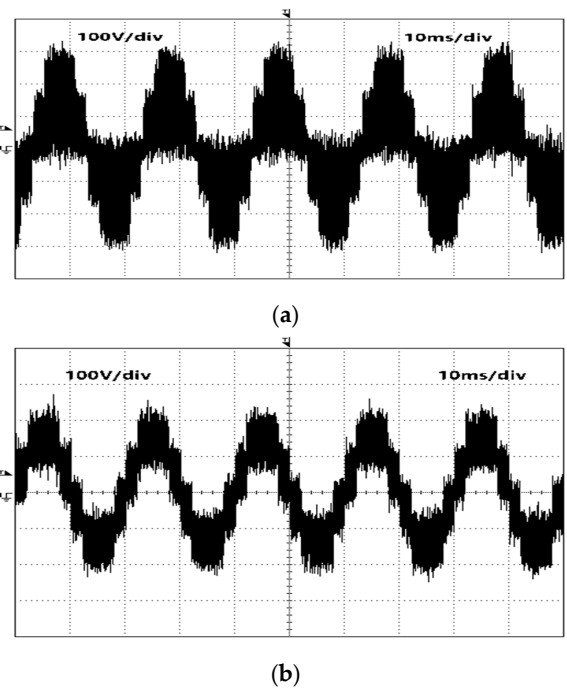

(**a**)

(**b**)

**Figure 21.** Experimental results of the load voltage for shoot-through of 0.05 and (**a**) existing modulation strategy or (**b**) newly proposed modulation strategy.

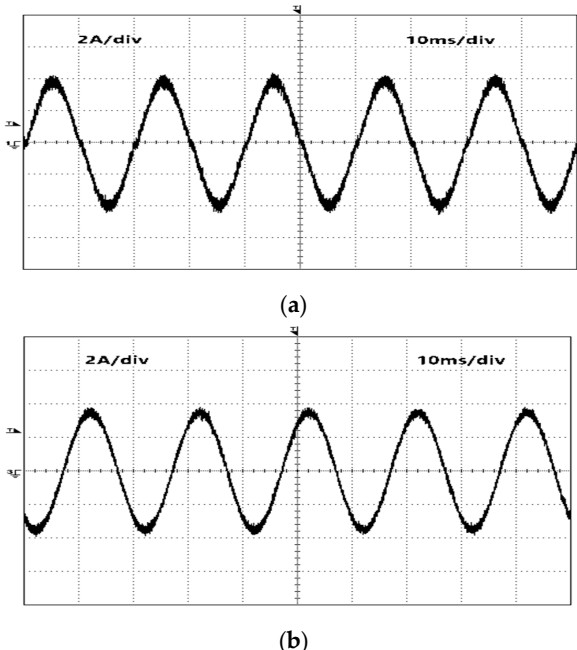

(**a**)

(**b**)

**Figure 22.** Experimental results of the load current for shoot-through of 0.05 and (**a**) existing modulation strategy or (**b**) newly proposed modulation strategy.

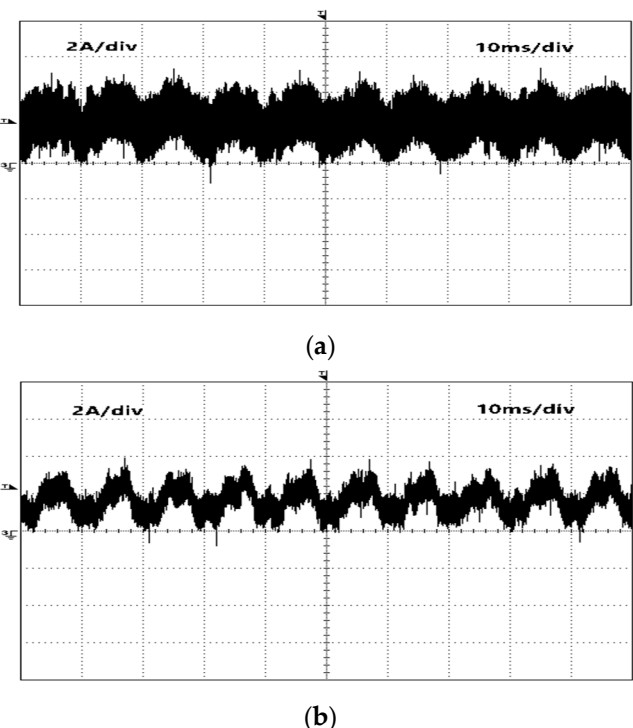

**Figure 23.** Experimental results of the converter input current for shoot-through of 0.05 and (**a**) existing modulation strategy or (**b**) newly proposed modulation strategy.

Similarly to the simulation tests, several laboratory tests were also performed for a different range of shoot-through duty cycle values to compare the behavior of the converter under the application of the two modulation strategies. Figures 24–26 show the AC load voltage, the AC load current, and the converter input current, respectively, considering a modulation index of 0.8 and an increased shoot-through value of 0.08. The obtained results also show that, although the load voltage and current amplitudes grow as expected, they continue to behave in the same way as in the previous experimental tests. It is also possible to see that, with a shoot-through of 0.08, the load current amplitude generated by the suggested new modulation strategy is roughly equivalent to that produced by the existing modulation strategy. Figure 24 also shows that the peak voltage level of the existing modulation (Figure 24a) is slightly higher when compared to the newly proposed modulation (Figure 24b) considering a shoot-through value of 0.08. The amplitude of the load voltage attained with the newly proposed modulation strategy is, however, less than the amplitude obtained with the existing modulation strategy with a shoot-through of 0.05 (Figure 21a), which agrees with the presented simulation results. However, for the same load current amplitude, less voltage amplitude is needed when using the newly proposed modulation strategy which can be seen as an advantage of the proposed solution, as well as the current TDH content reduction. The converter input current's decreased ripple when using the newly proposed modulation strategy also confirms the simulations results. This is also a clear advantage of using the newly proposed modulation strategy. In fact, the current ripple value obtained value with the classical modulation is around 69%, while the current ripple value obtained with the newly proposed modulation is around 42%. These values also confirm that the current ripple between the modulations is not constant, being lower for lower shoot-through duty cycles.

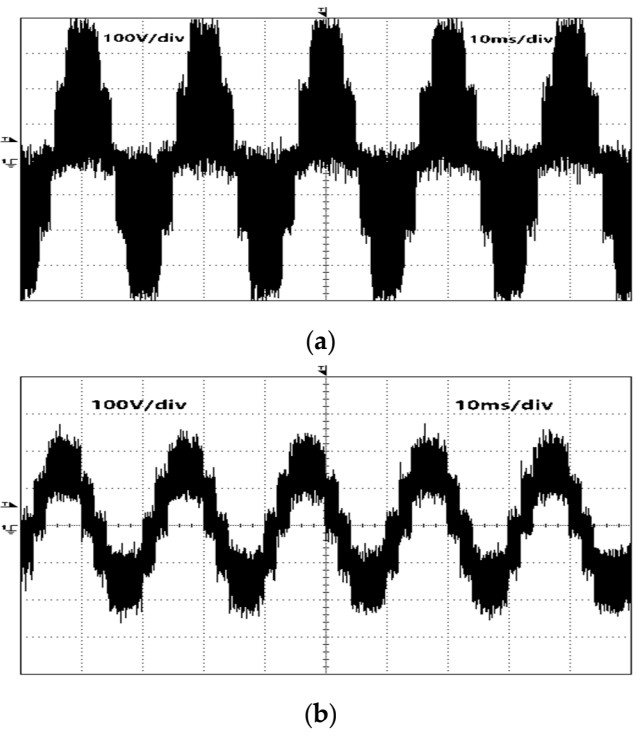

(**a**)

(**b**)

**Figure 24.** Experimental results of the load voltage for shoot-through of 0.08 and (**a**) existing modulation strategy or (**b**) newly proposed modulation strategy.

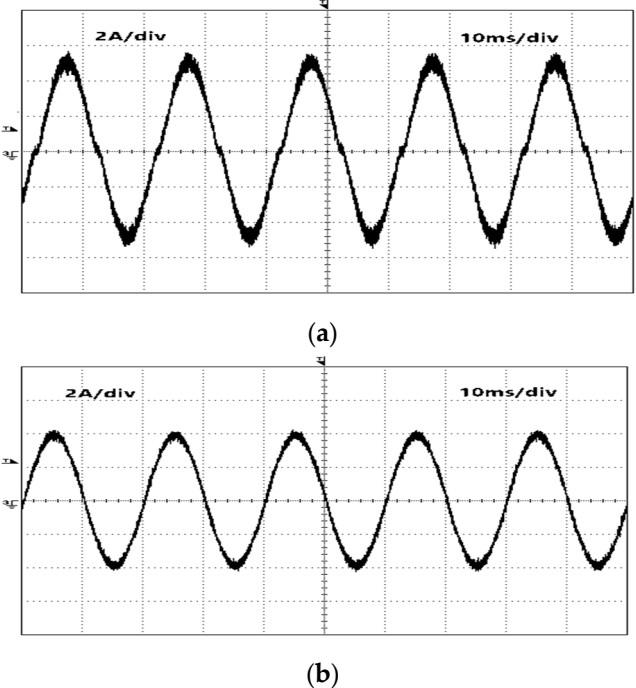

(**a**)

(**b**)

**Figure 25.** Experimental results of the load current for shoot-through of 0.08 and (**a**) existing modulation strategy or (**b**) newly proposed modulation strategy.

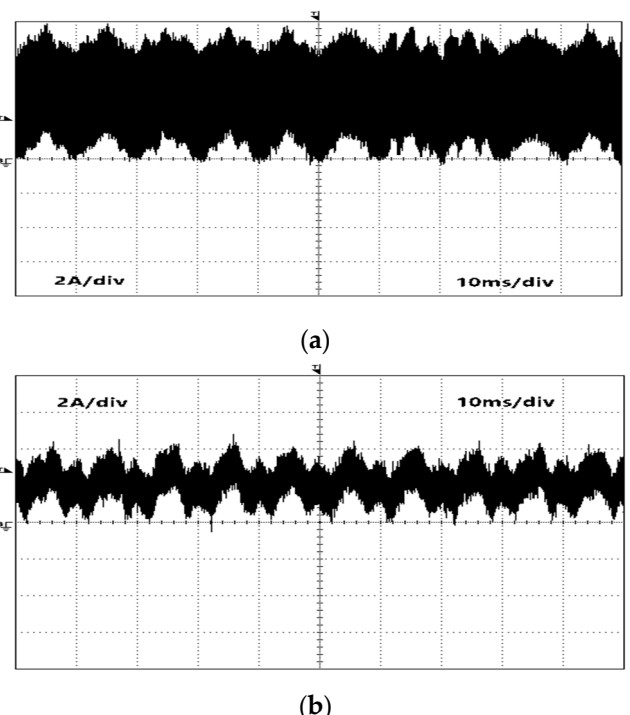

**Figure 26.** Experimental results of the converter input current for shoot-through of 0.08 and (**a**) existing modulation strategy or (**b**) newly proposed modulation strategy.

The harmonic distortion of the load waveforms produced by both modulation techniques was also obtained by experimental tests. This practical evaluation was performed for increasing the shoot-through duty cycle values. Figure 27 shows the results obtained for the voltage and current THD as function of the shoot-through duty cycle. The THD values were obtained using a Yokogawa oscilloscope with a 150 MHz bandwidth and 1 GS/s sampling. The values were then transposed to a spreadsheet in order to print the line graphs. This figure attests that, as shoot-through increases, the THD of the load voltage (THD-V) and current (THD-I) waveforms increase quickly when using the standard modulation strategy. This is not the case, however, with the newly proposed modulation strategy, as the THD stays reduced and nearly constant even as the shoot-through duty cycle increases. Experimental tests were also performed to evaluate the converter efficiency. Figure 28 shows the converter efficiency for a range of shoot-through duty cycle values. According to the obtained experimental results, the newly proposed modulation strategy improves efficiency by around 1%, as the newly proposed modulation technique reduces the number of switches that change state during transitions. The efficiency result presented in Figure 28 was calculated based on the measurement of the input power and output power, the ratio of which results in the efficiency. Additionally, to confirm this result, another method was also performed considering an analytical evaluation and calculation. The analytical calculation is based on the switching waveforms of all active and passive component voltages and currents. These waveforms were observed in the oscilloscope screen (stopped screen and observing a single period waveform for each device) using the $\Delta V$ (or $\Delta I$) and $\Delta T$ tools [40]. These tools are lines that show up in the oscilloscope screen to measure the voltage (or current) and time differences which help to calculate the respective areas. The comparison of both methods revealed that the solution of measuring the input and output power provides similar results when compared with the analytical evaluation.

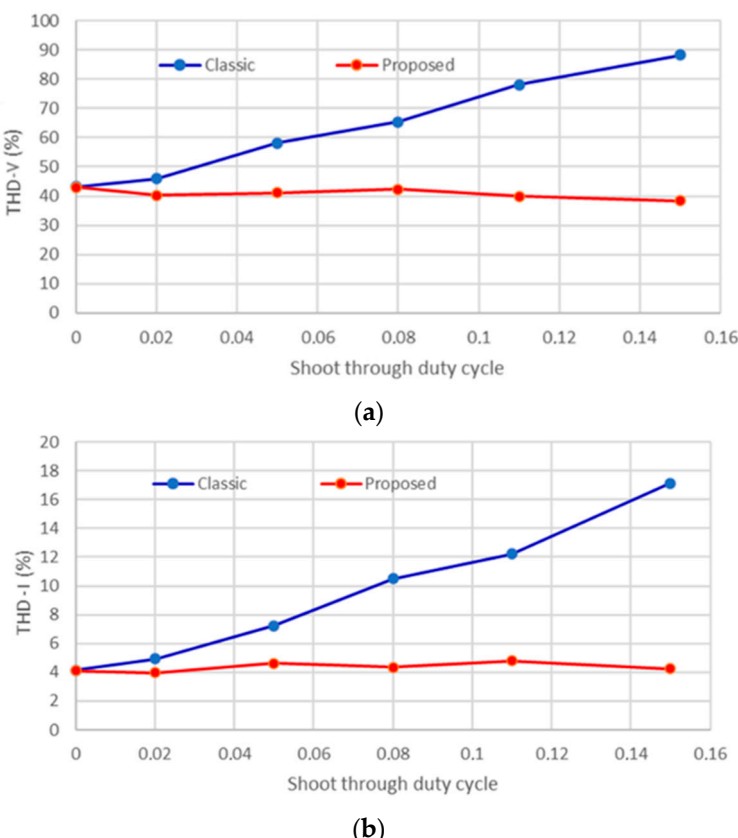

**Figure 27.** Experimental results of the THD function of the shoot-through duty cycle of the (**a**) load voltage (THD-V); (**b**) load current (THD-I).

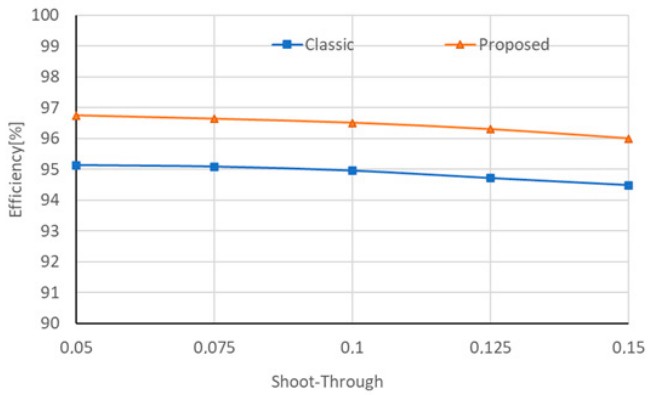

**Figure 28.** Experimental results of the efficiency function of the shoot-through duty cycle.

## 7. Conclusions

This work is focused on the improvement of the SPWM modulation strategy for the Single-Phase T-Type qZ Source Inverter. This converter is indicated for use in renewable generators, like photovoltaic applications, and, to achieve the necessary voltage level and enable the system to operate over a larger range of PV output voltage, the qZS network can be taken into consideration for PV voltage regulation. The proposed SPWM modulation strategy mitigates the single-phase T-Type qZS inverter's loss of performance, increased electromagnetic interference, and harmonic distortion caused by the shoot-through state in existing modulation approaches. In this context, this paper proposed a new modulation approach for the single-phase T-Type qZS to enhance the quality of the load voltage and current, while increasing the efficiency as a consequence of the reduction in the switching change states during transitions. To increase the output voltage and input current quality,

the newly proposed modulation avoided the shoot-through states connecting the upper and lower DC poles. Instead, when the shoot-through states need to be activated, the new modulation uses a new strategy, based on the upper and lower shoot-through states, with a reduction in the number of switches changing state, in order to avoid voltage decreases bigger than one voltage level. Simulation and experimental testing confirmed the prediction that the quality of the load voltage and current would improve, together with that of the input current. Prepared tests confirmed a reduction in the harmonic distortion in both the load voltage and the current, at the cost of a slight reduction in the converter boost capability, which is offset by the fact that lower peak voltages are needed to have the same output current. While in the existing modulation strategy the harmonic distortion increases rapidly as the shoot-through duty cycle increases, the newly proposed modulation strategy results show that harmonic distortion remained nearly constant as the shoot-through duty cycle increased. The simulation results showed that the newly proposed modulation presents and output voltage distortion of around 50% and an output current distortion of around 13%. Moreover, the newly proposed modulation strategy also reduced the ripple of the converter input current. In this case, an input current ripple reduction of around 27% was measured. Several tests revealed that the newly proposed modulation improves the converter efficiency around 1%. One aspect that was possible to see is that the comparative advantages are obtained at the cost of a lower converter voltage gain in the peak voltage. Future work is needed to quantify and compensate for this new modulation disadvantage, using the ratio of the first harmonic RMS output voltage to the RMS output current.

**Author Contributions:** Conceptualization, V.F.P. and A.C.; methodology, V.F.P., A.C. and D.F.; software, V.F.P. and C.R.-C.; validation, D.F. and A.C.; formal analysis, V.F.P. and E.R.-C.; investigation, V.F.P. and A.C.; resources, V.F.P., D.F. and A.C.; writing—original draft preparation, V.F.P. and A.C.; writing—review and editing, V.F.P. and J.F.S.; supervision, V.F.P. and J.F.S.; All authors have read and agreed to the published version of the manuscript.

**Funding:** This work was supported data underlying by national funds through FCT, Fundação para a Ciência e a Tecnologia, under project UIDB/50021/2020 (DOI:10.54499/UIDB/50021/2020) and UIDB/00066/2020.

**Data Availability Statement:** All the results are available as part of the article and no additional source data are required.

**Conflicts of Interest:** The authors declare no conflict of interest.

## Abbreviations

| | |
|---|---|
| VSI | Voltage Source Inverters |
| qZS | Quasi-Z Source |
| DC | Direct Current |
| AC | Alternating Current |
| SBC | Simple boost Control |
| MBS | Maximum Boost Control |
| MCBC | Maximum Constant Boost Control |
| NPC | Neutral Point Clamped |
| UST | Upper Shoot-Through |
| LST | Lower Shoot-Through |
| SVM | Space Vector Modulation |
| EMI | Electromagnetic Interference |
| PV | Photovoltaic Panels |
| $D_S$ | Duty Cycle |
| SPWM | Sinusoidal Pulse With Modulation |
| PWM | Pulse With Modulation |
| RMS | Root Mean Square |
| THD | Total Harmonic Distortion |
| $V_i$ | Input Voltage |

| V$_o$ | Output Voltage |
| V$_L$ | Inductor Voltage |
| V$_C$ | Capacitor Voltage |
| I$_o$ | Output Current |
| I$_L$ | Inductor Current |
| I$_C$ | Capacitor Current |
| S$_i$ | Semiconductor Switch |
| f | Switching Frequency |
| I$_U$ | Upper Current |
| I$_L$ | Lower Current |

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
