# Peer review of "Improved Carrier-Based Modulation for the Single-Phase T-Type qZ Source Inverter"

_electronics, doi:10.3390/electronics13061113_

Round 1
Reviewer 1 Report
Comments and Suggestions for Authors
This paper proposes a modified carrier-based control method for the T-Type single-phase quasi-Z inverter. The authors are encouraged to consider the following comments in their revised manuscript:
1. The abstract needs improvements. Please add more information about the proposed control method in the abstract and briefly discuss the findings demonstrating the importance of the paper’s contribution. The authors are suggested to follow the Introduction, Methods, Results, and Discussion (IMRaD) abstract writing style.
2. The research gaps are not clearly discussed in Section 1. Please clarify the need to propose the modified carrier-based control method based on the gaps in the literature (refs 1 to 28).
3. At the end of Section 1, the main contributions of this paper must be highlighted. A list of 2-3 bullet points must be added to clearly present, justify, and describe the contributions, features of the proposed method, the differences between the proposed method and the previous methods in the literature, and the main steps applied throughout the manuscript to verify the robustness of the proposed method. Mentioning that the method is validated simply by simulations and experiments is insufficient. This will strengthen the value of this paper.
4. After highlighting the main contributions of this paper (comment 3), another small paragraph must be added before starting with Section 2. This paragraph must present the remaining sections of this paper.
5. It is not clear which theory or equations used in Sections 3 and 4 are proposed by the authors or taken from previous references. For example, it seems that Figs 1 and 2 are taken from previous papers. Hence, please add the relevant references where it is necessary to clarify that the used information is taken from previous papers. Moreover, adding some remarks would also help the readers distinguish which information, figure, or theory is developed or proposed by the authors.
6. Please provide the rationale of selecting the parameters’ values given in Table 2.
7. Please discuss the disadvantages of the proposed control method and the suggestions to overcome them in the conclusion as a future work.

Comments on the Quality of English Language
Moderate editing of English language required
Author Response
Response to Reviewer 1 Comments
Point 1: This paper proposes a modified carrier-based control method for the T-Type single-phase quasi-Z inverter. The authors are encouraged to consider the following comments in their revised manuscript:
- The abstract needs improvements. Please add more information about the proposed control method in the abstract and briefly discuss the findings demonstrating the importance of the paper’s contribution. The authors are suggested to follow the Introduction, Methods, Results, and Discussion (IMRaD) abstract writing style.
Response 1: Thank you very much for your analysis of our work and your respective comments. You are completely right. In fact, this needs an important improvement. Thus, we improved the abstract in accordance with your suggestions. Thank you.
Point 2: 2. The research gaps are not clearly discussed in Section 1. Please clarify the need to propose the modified carrier-based control method based on the gaps in the literature (refs 1 to 28).
Response 2: You are completely right. Thus, in order to overcome this, we introduce a clarification of this in the Section 1 Introduction. Besides that, we also introduced a new Section 4 “Comparative study”, where this aspect is also pointed out.
Point 3: At the end of Section 1, the main contributions of this paper must be highlighted. A list of 2-3 bullet points must be added to clearly present, justify, and describe the contributions, features of the proposed method, the differences between the proposed method and the previous methods in the literature, and the main steps applied throughout the manuscript to verify the robustness of the proposed method. Mentioning that the method is validated simply by simulations and experiments is insufficient. This will strengthen the value of this paper
Response 3: This is a very important remark. In fact, this allows us to highlight the main contribution of the proposed modulation technique. At the end of Section 1, bullets points were introduced to highlight the main contributions of the paper. Thank you very much.
Point 4: 4. After highlighting the main contributions of this paper (comment 3), another small paragraph must be added before starting with Section 2. This paragraph must present the remaining sections of this paper.
Response 4: You are completely right. Thus, at the end of the Introduction, we added a paragraph to present the remaining sections of the paper.
Point 5: 5. It is not clear which theory or equations used in Sections 3 and 4 are proposed by the authors or taken from previous references. For example, it seems that Figs 1 and 2 are taken from previous papers. Hence, please add the relevant references where it is necessary to clarify that the used information is taken from previous papers. Moreover, adding some remarks would also help the readers distinguish which information, figure, or theory is developed or proposed by the authors
Response 5: This remark is timely. In the paper new version, we addressed this aspect. Thus, references to the Figs and theory from other authors were introduced. The paper main contributions are in Section 3. However, we started with a new sentence with a reference to clarify that existing modulation strategy. All other content in this section is developed by the authors. Thank you.
Point 6: Please provide the rationale of selecting the parameters’ values given in Table 2.
Response 6: This question is very pertinent. Indeed, the methodology that allows for determining the parameters of the converter was not presented. Thus, we included in the manuscript the design of the power converter components, including the power semiconductors stresses.
Point 7: Please discuss the disadvantages of the proposed control method and the suggestions to overcome them in the conclusion as a future work.
Response 7: You addressed a very important issue. In fact, this was not very clear. Thus, we complement the conclusions with quantified reductions and improvements in order to better clarify the advantages. On the other hand, it was also clarified the disadvantage of the proposed approach. Suggestions to overcome them in the conclusions were also introduced.
Reviewer 2 Report
Comments and Suggestions for Authors
The manuscript presents a modulation strategy for the multilevel quasi-Z single phase inverter. There are a number of points of particular significance to better this manuscript.
1. The title should be clear and informative and should reflect the aim and approach of the work. Your title is too long and should be reduced. A good title should not exceed 10 words (up to 12 if necessary).
2. According to Table 2, a switching frequency of only 5 kHz was used. The proposed work could be validated at a much higher frequency (at least 50 kHz).
3. How the components values in Table 2 are selected. A detailed step-by-step design methodology must be included in the paper. Also include detail about switch stress, efficiency, gain etc.
4. I noticed some redundant states, for the proposed strategy. They might affect the output voltage distortion. Please explain how the proposed PWM modulation strategy is more efficient comparatively to available techniques.
5. Provide a more comprehensive introduction to the topic. Include a brief overview of previous research in the area and contextualize the novelty and significance of your work. Identify, clear research gap?
6. What is the main difference between your proposed work and previous available modulation strategies? Provide a thorough comparison of the proposed work with existing techniques. Highlight the advantages and limitations of the proposed work? Include a separate section for comparison to show benefits of your work. It is suggested that the key features, such as voltage stress, current ripples, number of components and efficiency etc, are summarized in a table. It is important that the advantages of the proposed work with the existing solutions are clearly highlighted. Please quantify the improvement made compared to the state of art. Please include a loss distribution analysis to demonstrate where the losses are concentrated in the proposed circuit.
7. In line 95, Fig. 2 should be written as Figure 2. Please use the appropriate journal format. Also same type of formatting is needed in other sections.
8. I propose adding a list of abbreviations and symbols at the end of the paper before the list of references.
9. The figures 4, 6, and 9 should be improved. The text inside figures should be visible. Please improve the visibility of these figures.
10. In line 159, please write both Figures as Figure 7 (a) and (b). Also, make the desire correction at Line 183.
11. In the conclusion section, reiterate the main contributions and key findings of the research. Clearly state the significance of the proposed work and their potential applications. Discuss any future research directions or improvements that could be explored based on the current findings.
Author Response
Response to Reviewer 2 Comments
Point 1: The manuscript presents a modulation strategy for the multilevel quasi-Z single phase inverter. There are a number of points of particular significance to better this manuscript.
- The title should be clear and informative and should reflect the aim and approach of the work. Your title is too long and should be reduced. A good title should not exceed 10 words (up to 12 if necessary)
Response 1: Thank you very much for your analysis of our work and respective comments. The tittle has been reduced to 12 words while still reflecting the approach and main contribution of the work done.
Improved Carrier-Based Modulation for the Single-Phase T-Type qZ Source Inverter
Point 2: According to Table 2, a switching frequency of only 5 kHz was used. The proposed work could be validated at a much higher frequency (at least 50 kHz).
Response 2: Your question is very pertinent. For the lab prototype, we used existing IGBTs and magnetics in the lab, which are not appropriate for 50kHz. We did also simulation studies using much higher frequencies, and the obtained differences in efficiency are marginal, provided the right high-frequency semiconductors, capacitors and magnetics are used. In order to maintain the coherency between the simulation and experimental results, we preferred to maintain the 5kHz frequency in experiments to validate the work, as increasing the switching frequency to reasonable values will not change the conclusions of the work, provided the right components are used. Thank you for this remark.
Point 3: How the components values in Table 2 are selected. A detailed step-by-step design methodology must be included in the paper. Also include detail about switch stress, efficiency, gain etc.
Response 3: Thank you for this very important suggestion. We included in the manuscript the design of the power converter components, including the power semiconductors stresses. We also present in Figures 17 and 28 the efficiency of the converter.
Point 4: I noticed some redundant states, for the proposed strategy. They might affect the output voltage distortion. Please explain how the proposed PWM modulation strategy is more efficient comparatively to available techniques.
Response 4: Thank you for pointing out this aspect. In fact, there are redundant states. They were selected in order to keep unchanged the desired output voltage waveform, reducing in this way the THD of the output voltage and current. These redundant states are only used to minimize the number of transitions in the transistors. To clarify this, we introduced in the paper that the redundant states are used to reduce the number of switches that change state during transitions without affecting the output voltage distortion, while decreasing the switching losses.
Point 5: Provide a more comprehensive introduction to the topic. Include a brief overview of previous research in the area and contextualize the novelty and significance of your work. Identify, clear research gap?
Response 5: You are completely right, thank you. Thus, we introduce a clarification of this issue in the Section 1 Introduction. Besides, at the end of Section 1, we also introduced bullet points, in which we highlighted the main contributions of the paper.
Point 6: What is the main difference between your proposed work and previous available modulation strategies? Provide a thorough comparison of the proposed work with existing techniques. Highlight the advantages and limitations of the proposed work? Include a separate section for comparison to show benefits of your work. It is suggested that the key features, such as voltage stress, current ripples, number of components and efficiency etc, are summarized in a table. It is important that the advantages of the proposed work with the existing solutions are clearly highlighted. Please quantify the improvement made compared to the state of art. Please include a loss distribution analysis to demonstrate where the losses are concentrated in the proposed circuit.
Response 6: You addressed very important points to be improved. In this way, we introduced a new Section 4 “Comparative study” where the above points are considered. Another point analysed and added to the paper was the converter Power-Loss Breakdown for the proposed new modulation and existing ones. Through these results, it was confirmed that the switching losses and harmonic losses associated with the proposed new modulation strategy become smaller when compared to those presented by existing modulations.
Point 7: In line 95, Fig. 2 should be written as Figure 2. Please use the appropriate journal format. Also same type of formatting is needed in other sections.
Response 7: Thank you very much for noticing these lapses. Besides the change of this, we also did a full revision of the document and corrected other issues.
Point 8: I propose adding a list of abbreviations and symbols at the end of the paper before the list of references.
Response 8: Thank you for the suggestion. At the end of the paper we added an abbreviations section. Thank you very much.
Point 9: The figures 4, 6, and 9 should be improved. The text inside figures should be visible. Please improve the visibility of these figures.
Response 9: This is very well pointed out, thank you. Indeed, their quality regarding the text is low. Thus, new figures in which it was improved the text inside the figures we introduced. Thank you very much.
Point 10: In line 159, please write both Figures as Figure 7 (a) and (b). Also, make the desire correction at Line 183.
Response 10: Thank you very much for noticing these lapses In reality, the whole circuit is difficult to represent in just one figure, as it has two sub-circuits. Now the two sub-circuits are represented in the same figure, one sub-circuit to the left, the output sub-circuit in the right. We hope this is the right way.
Point 11: In the conclusion section, reiterate the main contributions and key findings of the research. Clearly state the significance of the proposed work and their potential applications. Discuss any future research directions or improvements that could be explored based on the current findings.
Response 11: Thank you very much for pointing this important issue. In accordance with this question, we strongly improved the conclusions, discussed the potential applications, reiterated the main contributions, and discussed future research directions.
Reviewer 3 Report
Comments and Suggestions for Authors
Comments:
1. Page 6 - Shoot-through and not thought
2. Add images of experimental setup and explanation of the setup
3. Figure 26 - Make plots in similar colors to the simulation
4. Which software was used for simulation? Add that in the appropriate section
5. Comment on the reduction in boost capability - How much is it reduced in terms of percentage or values (Quantification will help)
6. Please provide a justification on choosing the parameters in Section4. Simulation results - For example is it based on solar application or the reasons for choosing the parameters.
7. More context on application in introduction will help the readers understands its application
Overall good paper and good work by the researchers
Comments on the Quality of English Language
English is good with very few grammatical errors.
Author Response
Response to Reviewer 3 Comments
Point 1: 1. Page 6 - Shoot-through and not thought.
Response 1: First of all, we would like to express our gratitude for your extremely careful and precise analysis. We would also like to thank you for noticing this lapse. Besides the elimination of this typo, we also did a full revision of the document.
Point 2: 2. Add images of experimental setup and explanation of the setup.
Response 2: Thank you for this suggestion. The photograph of the lab prototype could be included in the original manuscript, but it was forgotten. Thanks to your suggestion, we included the lab prototype photograph (Figure 20) in the paper revised version.
Point 3: 3. Figure 26 - Make plots in similar colors to the simulation
Response 3: Yes, you are right. In fact, this is the correct way. Therefore, we changed the plot of figure 26 in order to maintain the similarity of the colors.
Point 4: 4. Which software was used for simulation? Add that in the appropriate section.
Response 4: Thanks to pointing this subject. Computer simulations were obtained using the well-known and established program Matlab/Simulink and its Power Systems Toolbox. This clarification is now written in the manuscript.
Point 5: 5. Comment on the reduction in boost capability - How much is it reduced in terms of percentage or values (Quantification will help).
Response 5: This is a very important comment. In the case study presented in the paper, the reduction is about 17% in the peak voltage. However, this reduction is not constant, but depends on the shoot-through duty cycle. Thus, for lower values of the shoot-through duty-cycle this reduction is also lower. Further work is needed to assess the effects on the output current and to mitigate the loss in the boost capability. These clarifications were introduced in the manuscript.
Point 6: 6. Please provide a justification on choosing the parameters in Section4. Simulation results - For example is it based on solar application or the reasons for choosing the parameters.
Response 6: This is very useful comment. These parameters were not chosen for a specific application, since the purpose is to test the proposed modulation approach. However, these parameters that can be used in photovoltaic systems in which serial connections between PV panels are adopted. These values are also compatible with 120 V RMS low-voltage grids. This clarification was introduced in the manuscript.
Point 7: 7. More context on application in introduction will help the readers understands its application.
Response 7: Thank you very much for pointing this. The single-phase T-Type qZS inverter is important in applications like renewable generators or in fuel cell applications. They are very well adapted for applications behaving as constant power sources. In fact, to achieve the necessary voltage level and enable the system to operate over a larger range of PV output voltage, a QZS network is taken into consideration for PV voltage regulation and AC current injection. With the proposed new modulation and the reduction in the number of switches changing states, the single-phase T-Type qZS inverter is a more dependable and efficient system. To clarify the readers, this topic was introduced in the introduction, as well as new references.
Point 8: Overall good paper and good work by the researchers.
Response 8: We would like to thank you very much. We are very happy to see our work appreciated by you.
Reviewer 4 Report
Comments and Suggestions for Authors
The paper is nicely written and easy to understand. However, it could be improved by having more references, as the topic is widely discussed in the literature. Also, many other papers propose modified carrier-based control methods for the T-type single-phase quasi-Z inverter topology. However, the authors do not discuss/compare their method with other "modified" methods.
In addition, I have the following comments that must be addressed:
1. Figure 6 and Figure 8 have the same titles. Same for Figure 7 and Figure 10. Please correct and align with the main text.
2. Section 4 argues that the current ripple is reduced by the proposed new modulation technique. Please give the value of the current ripple, as the figures alone are not sufficient to validate this conclusion.
3. Section 4, lines 229-232, states that the load voltage peak is smaller for the new proposed modulation technique, and Figure 15b and Figure 12a are used as a proof. However, the indicated figures show current actually, not voltages, so please correct. Also, please give the numerical value of the load voltage peak to allow for a direct comparison. Finally, please comment why it is an advantage for the load voltage peak to be smaller.
4. Line 266-268 -> it is claimed that the converter efficiency has been tested/measured, but actually this text is still part of Section 4, which covers the Simulation results. So please correct the main text (actually Figure 18 clearly states that the efficiency is obtained by simulations, not be testing or measurements).
5. Please indicate in Figure 17 on the label of the 2 sub-figures what THD is shown, i.e. THD-V for sub-figure a) and THD-I for sub-figure b).
6. Same comment as Comment 5 for Figure 25.
7. Please give the numerical values of the measured current ripples measured experimentally (Section 5).
8. How did you measured the efficiency of the converter for the conventional, respectively the new proposed modulation techniques (Figure 25). Please explain.
9. Please include a photography of the experimental setup.
Author Response
Response to Reviewer 4 Comments
Point 1: The paper is nicely written and easy to understand. However, it could be improved by having more references, as the topic is widely discussed in the literature. Also, many other papers propose modified carrier-based control methods for the T-type single-phase quasi-Z inverter topology. However, the authors do not discuss/compare their method with other "modified" methods.
In addition, I have the following comments that must be addressed:
- Figure 6 and Figure 8 have the same titles. Same for Figure 7 and Figure 10. Please correct and align with the main text.
Response 1: Thank you very much for your analysis of our work and respective comments. You are completely right. The figure captions 6, 7, 8 and 10 are now corrected and aligned. Thank you.
Point 2: 2. Section 4 argues that the current ripple is reduced by the proposed new modulation technique. Please give the value of the current ripple, as the figures alone are not sufficient to validate this conclusion.
Response 2: Thank you very much for this helpful comment. You are completely right since the scale of the figures is not sufficient to validate this conclusion regarding the current waveforms. Thus, to clarify this, we introduced in the manuscript values associated with the current ripples. In fact, this allowed to clarify this issue.
Point 3: 3. Section 4, lines 229-232, states that the load voltage peak is smaller for the new proposed modulation technique, and Figure 15b and Figure 12a are used as a proof. However, the indicated figures show current actually, not voltages, so please correct. Also, please give the numerical value of the load voltage peak to allow for a direct comparison. Finally, please comment why it is an advantage for the load voltage peak to be smaller.
Response 3: Thank you very much for noting this mistake. Figures are now correct with the appropriate label (voltage and current) and the correction was made about the figures. The information about the peak voltage values and importance of smaller load voltage peak is now presented in section 5 (section 4 of the first submission).
Point 4: Line 266-268 -> it is claimed that the converter efficiency has been tested/measured, but actually this text is still part of Section 4, which covers the Simulation results. So please correct the main text (actually Figure 18 clearly states that the efficiency is obtained by simulations, not be testing or measurements).
Response 4: Thank you very much for noting this mistake. It is in fact simulation and not experimental which we stated by mistake. This aspect is now correct in this revised version.
Point 5: Please indicate in Figure 17 on the label of the 2 sub-figures what THD is shown, i.e. THD-V for sub-figure a) and THD-I for sub-figure b).
Response 5: Thank you very much for this helpful suggestion. The label of these two sub-figures (Fig.17) is now updated according to your suggestion. This aspect is now correct in this revised version.
Point 6: Same comment as Comment 5 for Figure 25.
Response 6: Thank you very much for this helpful suggestion. The label of these two sub-figures (Fig.25) is now updated according to your suggestion. This aspect is now correct in this revised version.
Point 7: Please give the numerical values of the measured current ripples measured experimentally (Section 5).
Response 7: Thank you very much for this helpful suggestion which will improve surely the quality of the paper. Thus, we introduced in the manuscript values associated with the current ripples. In fact, this allowed to clarify this issue.
Point 8: How did you measured the efficiency of the converter for the conventional, respectively the new proposed modulation techniques (Figure 25). Please explain.
Response 8: This is a very important remark. The adopted solution to calculate the experimental efficiency was based on the measurement of the input power and output power and then efficiency calculation. Additionally, other method to confirm the obtained results it was also performed considering an analytical evaluation and calculation. The analytical calculation is based on the switching waveforms of all active and passive component voltages and currents. This method consists of measuring separately the effective area of each section of both voltage and current for each device and multiply the areas (similar to perform a mathematical integration) to achieve the power losses of each device and sum all the results. These waveforms were observed in the oscilloscope screen (stopped screen and observing a single period waveform for each device) using the DV and DT tools. The comparison of both methods revealed that the solution of measuring the input and output power provide similar results which is acceptable. This clarification was introduced in the manuscript. Thank you.
Round 2
Reviewer 1 Report
Comments and Suggestions for Authors
The authors have properly responded to all comments. Thus, I have no further issues.
Still, I suggest the authors to further improve the manuscript's language, especially the modified parts in this version. Also, please pay attention to the journal's format regarding tables, figures, captions, references, and text.
Comments on the Quality of English Language
Further improvements are required especially in the modified parts of this version.
Author Response
The authors have properly responded to all comments. Thus, I have no further issues.
Still, I suggest the authors to further improve the manuscript's language, especially the modified parts in this version. Also, please pay attention to the journal's format regarding tables, figures, captions, references, and text.
Comments on the Quality of English Language
Further improvements are required especially in the modified parts of this version.
Response : Thank you very much for your analysis of our work and your respective comments. We also like to thank you for your very positive feedback.
Regarding the issues that you pointed out, you are completely right, and we did a very carefully revision of the manuscript.
Reviewer 2 Report
Comments and Suggestions for Authors
The quality of the manuscript increased significantly. I am satisfied from the revised version.
Author Response
The quality of the manuscript increased significantly. I am satisfied from the revised version.
Response : Thank you very much for your analysis of our work and your respective comments. We also like to thank you for your very positive feedback.
Reviewer 4 Report
Comments and Suggestions for Authors
All of my previous comments have been addressed properly by the authors. However, I kindly ask, in relation to comment 8, that the authors include the explanation of what DV and DT acronyms mean (line 553)
Comments on the Quality of English Language
Please prof-read your paper, with focus on the text added after the 1st review round, before the final version. For example, have a look at, and rephrase, the sentences on lines 312-313, 363-364, 546-549, as these are not so clear and in some instances the verb is missing.
Author Response
All of my previous comments have been addressed properly by the authors. However, I kindly ask, in relation to comment 8, that the authors include the explanation of what DV and DT acronyms mean (line 553)
Comments on the Quality of English Language
Please prof-read your paper, with focus on the text added after the 1st review round, before the final version. For example, have a look at, and rephrase, the sentences on lines 312-313, 363-364, 546-549, as these are not so clear and in some instances the verb is missing.
Response : Thank you very much for your analysis of our work and your respective comments. We also like to thank you for your very positive feedback.
Regarding the issues that you pointed out, you are completely right. In this new version we addressed those issues and we did a very carefully revision of the manuscript.